# Determinants of farmers' decision-making behavior for black soil protection in Northeast China—A case of Jilin Province*

Xu Sun , Rui Yan, Ling Zhao *, Yunxian Yan*

College of Economics and Management; Jilin Agricultural University, Changchun, Jilin, China

* 1679182@qq.com (ZL); yanyunxian@126.com (YY)

## Abstract

Northeast black soil area is one of the world's four major pieces of black soil area, which has a pivotal position in ensuring national food security. But in recent years it degraded, it is urgent to protect it. Farmers are the main body directly involved in black soil protection, and farmers' response, that is, perceived status, directly reflects their recognition, acceptance and active cooperation with black soil protection, which is an important factor affecting their willingness to protect black soil. Therefore, based on the survey data of 486 farmers in black soil area of Jilin Province, this study studied farmers' intention choice and behavior response based on the theory of planned behavior.The empirical results showed that the farmers' black soil protection behavior followed the logic of Farmer's Cognition-Behavioral Intention-Behavioral and farmers' protection behavior response is determined by their behavior intention, behavior attitude, subjective norms and perceived behavioral control. Specifically, in the Perceived Behavioral Control→Behavioral Intention→Behavior path, there were both significant direct and indirect effects between variables, while in the Attitude toward Behavior→Behavioral Intention→Behavior path there were only significant indirect effects. There are significant correlations among farmers' behavioral attitudes, subjective norms, and perceptual behavioral control. In order to protect the black soil successfully, the farmers' willingness and income should be paid enough attention to and the central and local governments should publicize and train rural carders, farmers and other stakeholders.

## 1. Introduction

As one of the three black soil areas in the Northern Hemisphere, Northeast China is an important area for grain production and the largest commercial grain production base in China. Farmland is an important input of grain production. Farmland security is an important guarantee and foundation for food security [1–4]. For over 20 years, China's arable land has shown a trend of increasing first and then decreasing [5],

**Data availability statement:** All relevant data are within the paper and its Supporting information files.

**Funding:** This work was supported by The Education Department of Jilin Province (JJKH20250609BS). The funders had no role in study design, data collection and analysis, decision to publish, or preparation of the manuscript.

**Competing interests:** The authors have declared that no competing interests exist.

China's arable land is 1.28 million km2, accounting for only 13% of the country's total land area in 2021 [4]. Sustainable utilization of arable land resources is a strategy to ensure national food security and social stability [6,7], farmland must be "protected like giant pandas". Northeast Black Soil Protection Planning Outline pointed that Northeastern black soil is one of the types of cultivated land with strong fertility and the ballast stone to guarantee China's food security and the supply of high-quality agricultural products [8]. To put it in another way, it is the "panda" in cultivated land. The black soil in Northeast China refers to a type of land with a surface layer of black or dark black humus. According to the Chinese soil genetic classification, the main soil types include black soil, chernozem, white soil, meadow soil, dark brown soil and brown soil [9]. It takes 200–400 years to generate 1 cm of black soil.

However, due to the long-term high-intensity use of black soil but lack of scientific conservation, and the black soil is faced with issues such as overdraft of soil fertility, reduction of soil organic matter content and damage to ecological functions, which resulted in the black soil layer becoming "thin", "lean" and "hard" [10]. "Thin" refers to the surface of the soil disappearing at a rate of 0.3–1 cm per year, and the overall resilience of cultivated land ecosystem is low [11]. "Lean" is associated with the soil's quality. The long-term intensive use, coupled with soil erosion, has led to a gradual decline in the natural fertility of the black soil year by year [12]. "Hard" is related to the high intensity of ploughing. This includes a significant decrease in the organic matter content in the topsoil, shallower plow layers, harder plow pans, and severe degradation of the soil's physical, chemical properties, and ecological functions. As a result, the black soil region in Northeast China has gradually transformed from an "ecological functional area" into an "ecologically fragile area [13]." This has seriously affected the sustainable development of agriculture in Northeast China. There is a risk that the role of the black soil in Northeast China as a foundation for the country's food production capacity may be undermined. Black soil resource security and agricultural ecological security are faced with severe challenges, and some studies have shown that the adoption of favorable cropland quality improvement and farmland ecological protection is conducive to the increase of farmers' welfare [14]. Therefore, it is critical to clarify the main determinants of the willingness and behavior of farmers to protect black soil and guide farmers to form the internal consciousness of black soil protection in order to improve the comprehensive production capacity of food.

The current literature on black soil protection have mainly focused on the scope of typical black soil areas [15], the physicochemical properties of black soil [16], the causes of black soil degradation and governance [17], conservation tillage techniques [18], and rule of law research [19] etc. The United States has had a significant impact on cultivated land area through CRP, with each plot having an associated environmental benefit index (EBI) [20], and Ukraine uses shelterbelt forests to reduce soil erosion and improve crop yields to protect cultivated land [21].It can be seen that the existing research has provided a useful reference for the promotion of black soil protection technology and policy system. Although conservation tillage in northeast China has made remarkable achievements [18], it also faces outstanding problems such as inadequate policy implementation and unsound regulatory mechanisms [19].

In particular, relevant studies have proved that farmers' lack of awareness will hinder the promotion and application of technology [17], and explored the impact mechanism of the factors on farmers' behavior, such as the background conditions of cultivated land, family structure, livelihood types, farmers' age, education level, cultural identity, social networks, government propaganda, subsidies and support and so on [22]. The existing literature revealed the mechanisms influencing farmers' willingness to protect black soil and the interactions among variables, but these investigations remain insufficient in several aspects, for instance, there are notable limitations in the pathways through which these factors influence relevant indicators and behavioral intentions. In terms of methodology, traditional linear equation models cannot directly observe or simultaneously deal with the relationship between certain indicators and latent variables.

As one of the most effective and widely used conceptual frameworks for analyzing attitudes and behaviors [23], planned behavior theory can significantly improve the predictive and explanatory power of behavior. Due to the close relationship between agricultural decision-making and social psychology, many agricultural economists and social psychologists apply the TPB to the field of agricultural research [24]. Some studies used the TPB to explain the adoption of water and soil erosion control measures by Belgian farmers [25], in combination with structural equation modeling using the five elements of behavioral attitudes, subjective norms, perceptual behavioral control, behavioral willingness, and behavioral [26,27]. Other studies modified the pre-latent variables and analyze the mechanism of farmers' behavior, which is the most common research application of planned behavior theory [28–30]. By deconstructing the planned behavior theory [31–34], taking behavior and attitude as intermediate variables [35] and adding six pre-variables such as perceived usefulness, perceived risk [36,37], peer influence [38], superior influence [28,34], self-efficacy [39] and convenience conditions, this paper analyzes the subjective decision-making factors that affect farmers' willingness to protect farmland, and finds that the biggest driving force for farmers to protect farmland is economic benefits [14], and farmers' trust in peers is higher than that in the village collective [40]. The existing literature has conducted in-depth research on farmers' cognition and behavior processes, such as the impact of environmental cognition and social cognition on farmland protection behavior [41], the influence of value cognition on farmland protection [42], and the effect of ecological cognition on farmland protection [43]. The research results indicate that there is an important influence mechanism between farmers' cognition and behavioral decision-making. Generally speaking, there is a lack of empirical research on farmers' black soil protection behavior.

Therefore, this paper, from a micro-farmer perspective, utilizes the TPB to construct an analytical framework of *farmer cognition-behavioral intention-behavioral* with a particular emphasis on the psychological factors of farmers, the study employs a structural equation model to systematically analyze the intrinsic relationships between observed variables and latent variables, as well as among various latent variables. This study aims at farmers' black soil protection behavior. What are farmers' intentional choices and behavioral responses? How to improve farmers' willingness to participate and active action?

The marginal contribution of this paper is the following three aspects. Firstly, this paper expands the existing research framework by constructing the theoretical framework of farmers' black soil protection behavior, identifying the quantitative indicators or proxy variables of black soil protection and evaluating the influence of farmers' black soil protection behavior. It enriches the theoretical framework system of black soil protection behavior and enhances the explanatory power of farmers' black soil protection behavior,. Secondly, this paper is done the analysis from the perspective of farmers, there are few decisions on farmers' black soil protection behavior at the micro level, and the existing research at the micro level mainly focuses on the influence of objective factors on farmers' black soil protection behavior, and the research on the decision-making mechanism of farmers' black soil protection behavior from subjective psychological cognitive factors is relatively rare. focusing on the influencing factors of farmers' black soil protection behavior, and exploring the design of black soil protection system, whose results contribute to the sustainable development of agriculture. Thirdly, we will then explore the design of a farmer-based black soil protection system that will contribute to the policy-making by the government. It is worth noting that this study mainly explores the logic of farmers' participation in black soil protection behavior,

but does not further analyze the regional differences of farmers' black soil protection behavior in different regions. Therefore, the mechanism constructed to promote farmers' black soil protection behavior may not fully consider the differences of farmers in resource utilization, policy response, production and management methods, cultural traditions and protection cognition in different regions. Future research needs to fully consider regional differences in order to obtain more accurate and comprehensive research results.

This paper is arranged as followed. Section 2 introduces the research theory-TPB, theoretical framework is constructed and hypotheses are put forward. Section 3 describes the material and methods, the study area is mapped, the data sources are identified, the descriptive analysis of the samples are demonstrated and the econometric model is set up. Results are explained in Section 4, it also covers the reliability and validity of the data. Section 5 concludes the paper with policy implication.

## 2. Theoretical framework and hypotheses

The TPB was developed by Ajzen [44] as an extension of the Theory of Rational Behavior to predict and explain human behavior in specific contexts in 1985. TPB is an attitude-behavior relationship theory, behavior attitude, subjective norms and perceived behavior control are the main structures to help predict behavior intention, which in turn affects and predicts people's behavior. According to TPB, personality traits and behavioral attitudes influence human behavior [45], intention is the most important predictor of behavior and is related to an individual's motivation or willingness to put in the effort to perform the behavior [31,46]. Some studies have suggested that people tend to conform to subjective norms due to fear of social rejection [31]. Thus, valuing others' opinions should boost farmers' intention to implement this behavior [47]. Perceived behavioral control reflects an individual's view of how easy or difficult it is to perform a behavior, and it is linked to the availability of facilitating conditions, often termed situational constraints [31,46].

According to TPB, behavioral attitude, subjective norm, and perceived behavioral control are the three major components of the theory. Behavioral attitude refers to an individual's positive or negative feelings towards a specific behavior; subjective norm refers to the external pressure an individual feels regarding whether to take a certain action; perceived behavioral control refers to the strength of an individual's belief in their ability to perform a behavior, which increases when they have more resources, information, or opportunities and fewer anticipated obstacles [48]. The essence of the TPB suggests that while behavioral attitude, subjective norm, and perceived behavioral control can be conceptually distinguished, at certain moments they are both independent of each other and pairwise related. Moreover, perceived behavioral control can directly influence or predict the occurrence of behavior. At the same time, behavioral attitude, subjective norm, and perceived behavioral control can influence human behavioral intention, which in turn affects the ultimate behavior of individuals, with behavioral intention being a key variable in subjective behavior. Behavioral intention refers to the psychological tendencies, willingness to cooperate, or plans that people have before engaging in a specific activity [49].

Consequently, this study employs behavioral attitude, subjective norm, perceived behavioral control, and behavioral intention as the four key independent variables to examine their impact on farmers' decision-making processes regarding the conservation of black soil.

### 2.1. The impact of behavioral attitude on farmers' black soil conservation behavior

Behavioral attitude can be understood as the positive or negative views that farmers hold towards specific aspects of black soil conservation. Expected benefits often explain farmers' attitudes towards black soil conservation behavior, encompassing both the direct benefits of conservation actions and indirect benefits such as enhanced black soil fertility, improved food security, and environmental betterment. Therefore, this paper measures farmers' behavioral attitudes across three dimensions: economic benefits, social benefits, and ecological benefits [50]. Implementing economic compensation programs for farmers at the national level is conducive to the sustainable development of cultivated land protection, particularly in countries like China, where arable land resources are scarce [51]. Therefore, "economic compensation

income satisfaction (AB1)" is used to represent the economic benefit of black soil protection behavior. Additionally, national food security encompasses quantity, quality, and structural security, necessitating not only sufficient cultivated land resources but also, more importantly, healthy cultivated land soil [52]. If farmers believe that food security is very important, they will be more willing to engage in black soil conservation behaviors; conversely, if they do not, they will not carry out black soil protection. Therefore, using "awareness of food security (AB2)" as an observed variable represents the social benefits brought about by black soil conservation behaviors. Research has indicated that the more positive farmers' perceptions of ecological utilization are, the more willing they are to protect cultivated land. In other words, if farmers recognize that black soil conservation behaviors enhance the ecological value of black soil, they will be more inclined to engage in black soil protection. Therefore, "awareness of ecological and environmental benefits (AB3)" is used to represent the ecological benefits brought about by black soil conservation behaviors.

Hypothesis One(H1) is proposed: In farmers' black soil protection behavior, attitude toward behavior (AB) has a positive effect on behavioral intention (BI).

## 2.2. The influence of subjective norms on farmers' black soil protection behaviors

Subjective norms can be understood as the external pressures that farmers perceive when making decisions about black soil protection behaviors. The groups that influence farmers' behavioral decisions typically include family members, neighbors, rural collective economic organizations, and the government [53]. The effectiveness of the government's propaganda and interpretation of relevant policies is a crucial factor in whether farmers choose to protect black soil [54], and neighborhood communication plays a significant role in the adoption of conservation farming practices by farmers [55]. Therefore, if the government has sufficient credibility in the minds of farmers, they will be more willing to respond to the government's call for black soil protection. If farmers have a high degree of agreement with the opinions of their relatives and friends, and their relatives and friends believe that the protection of black soil is very important, then they will be more willing to engage in black soil protection. In summary, this paper uses "advocacy for participation by relatives and friends(SN1)"、"advocacy for participation by rural collective economic organizations(SN2)"、"advocacy for participation by villagers(SN3)" and "the intensity of dissemination for black soil protection(SN4)" to represent the subjective norms that farmers are subject to.

Hypothesis Two(H2) is put forward: In farmers' black soil protection behavior, subjective norm (SN) has a positive effect on behavioral intention (BI).

## 2.3. The impact of behavioral perception control on the black soil protection behaviors of farmers

Perceptual behavior control can be understood as the extent to which the resources or information that farmers possess can influence their decision-making regarding black soil protection behaviors. Indeed, farmers' behavioral decisions are influenced not only by their own intentions and the intentions of others but also by many non-volitional factors [40]. These non-volitional factors can include environmental conditions, economic constraints, technological access, and institutional policies, which can significantly impact the ability and willingness of farmers to engage in black soil protection behaviors. These external factors often limit the degree to which personal intentions can be translated into actual behavior. The factors that control farmers include both intrinsic behavioral capabilities and extrinsic resources and support. Some scholars have summarized these two aspects of control factors as perceived control strength and control beliefs [56]. Generally speaking, the higher the level of education of farmers, the stronger their perceived behavioral control ability [57]. However, in actual research, the education level of farmers is generally low, with little differentiation, and it is difficult to represent the differences in their knowledge levels based on life experiences. Therefore, this paper uses farmers' "cognitive level of the black soil protection process (PBC1)" to represent perceived control strength; control beliefs include farmers' "cognitive level of black soil protection measures (PBC2)" as well as their "cognitive level of investment and use of funds for black soil protection (PBC3)." Theoretically, the stronger farmers' perceived control strength and control beliefs regarding black soil protection behavior, the higher their intention to participate and the more active their actual actions will be.

 

Hypotheses Three(H3) and Four(H4) are formed: In farmers' black soil protection behavior, perceived behavioral control (PBC) has a positive effect on behavioral intention (BI) of black soil protection; In farmers' black soil protection behavior, perceived behavioral control (PBC) has a positive effect on behavior (BE) of black soil protection.

Additionally, there may be pairwise correlations among behavioral attitude, subjective norms, and perceived behavioral control.

Hypothesis Five(H5) is proposed: There are pairwise correlations among farmers' behavioral attitude, subjective norms, and perceived behavioral control in the context of black soil protection behavior.

### 2.4. The impact of behavioral intention on farmers' black soil protection behavior

According to the TPB, the stronger of the intention by farmers to protect black soil, the more active their actual protective behaviors will be. Behavioral intention is influenced by behavioral attitudes, subjective norms, and perceived behavioral control, which in turn affect behavior, effectively acting as a mediating role. Funds, technology, and land are essential production factors for the entities engaged in agricultural production on black soil, and the supply of these production factors directly affects the enthusiasm of producers. At the same time, the acquisition of funds and technology also impacts the endowment of black soil resources [50]. Therefore, farmers' behavioral intentions are mainly reflected in "actively promoting black soil protection (BI1)","being willing to continuously protect black soil (BI2)","being willing to invest funds in black soil protection(BI3)" and "being willing to learn conservation tillage techniques(BI4)" The behaviors are mainly manifested in "strictly adhering to relevant protection regulations during the use of black soil(BE1)" "adopting conservation tillage measures such as crop rotation and fallow in black soil cultivation(BE2)" and "actively participating in black soil protection projects or training(BE3)".

Hypothesis Six(H6) is put forward: In the decision-making of farmers' black soil protection behavior, farmers' behavioral intention (BI) has a positive effect on behavior (BE).

According to the literature review and hypothesis development, we construct a conceptual model as presented in Fig 1.

## 3. Materials and methods

### 3.1. Study area

The Northeast China Black Soil Region is primarily located in Heilongjiang, Jilin, and Liaoning provinces, as well as the Inner Mongolia Autonomous Region, specifically in the "four eastern leagues" [Hulunbuir League (now Hulunbuir City), Xing'an League, Tongliao City (formerly Zherimen League), and Chifeng City (formerly Zhaowuda League)], the total area of the Northeast China Black Soil Region is 1.09 million square kilometers, accounting for approximately 12% of the global Black Soil Region's total area. Jilin is a major agricultural province and a significant grain-producing region, serving as an important commodity grain production base in China. Jilin promulgated and implemented China's first local regulations on black land protection, developed and produced the country's first tractor-trailed heavy-duty no-till planter, and saw the implementation area of conservation tillage increase from 4.5 million mu (each mu equals approximately 666.67 square meters) in 2015 to 18.52 million mu. The "Lishu model" of tillage conservation, recognized as the largest scale initiative in the country, received commendation from General Secretary Xi Jinping. The findings of the third national land survey reveal that Jilin province possesses 98.11 million mu of valuable black soil arable land, representing a substantial 87% of the province's total arable land area. Furthermore, the adoption of the 'Lishu model' in the pilot field in Jilin province has led to a remarkable increase in soil water content, rising from 20% to 40% over the past decade. This innovative approach has also significantly reduced soil loss by 80%. Additionally, the organic matter content in the soil within the plough layer, ranging from 0 to 20 cm in depth, has seen a substantial increase of nearly 13%. As a result of these improvements, in 2024, the area dedicated to grain cultivation is expected to reach 87.807 million mu, with grain yields projected to hit 971.7 catties per mu, thereby securing Jilin province's position as the top grain-producing region in the country.

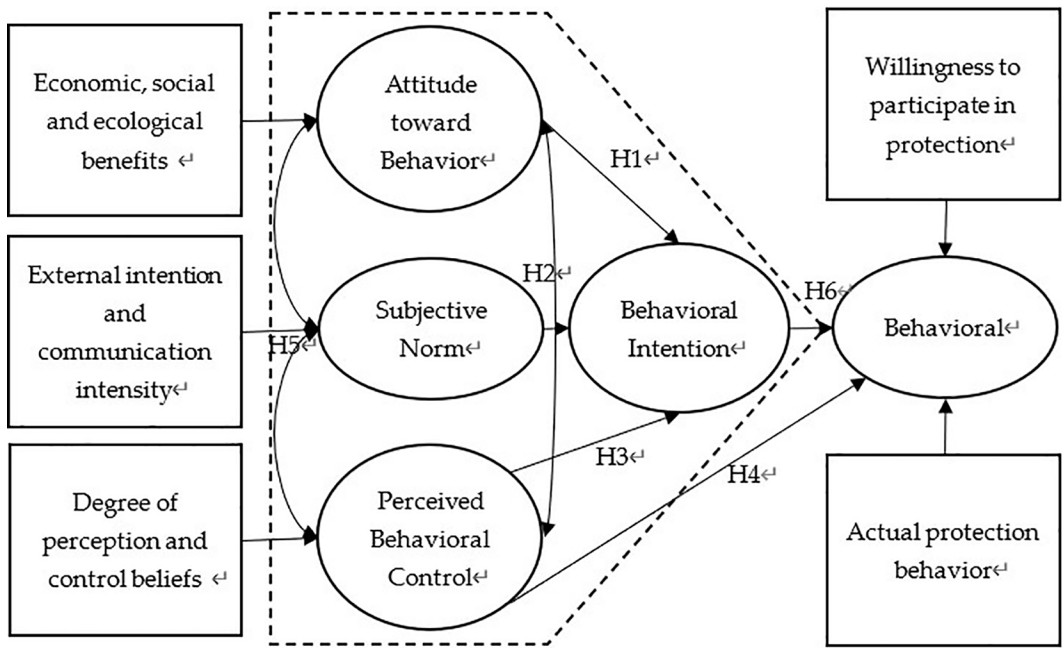

**Fig 1. Decision-making model of farmers' black soil protection behavior based on planning behavior theory.** Note: The constructs in the black dotted box is the TPB.

Jilin Province is located in the geometric center of Northeast Asia consisting of Japan, Russia, North Korea, South Korea, Mongolia and northeastern China. It spans between 121°38'-131°19' E and 40°50 '-46°19' N. Jilin Province as a typical representative of the main grain-producing areas in Northeast China, is located in the Songnen Plain of China, with good quality of arable land and hydrological and climatic conditions, and is an important province for China's grain production, especially corn production. Therefore, the production and management behavior of Jilin farmers and the protection of arable land quality are typical and representative of the province. It belongs to the temperate continental monsoon climate, four distinct seasons, rain and heat in the same season. The black soil area is about 1.1 million hectares, and the black soil cultivated land is about 832,000 hectares, accounting for 15.6% of the province's cultivated land area, and the grain output in the black soil area accounts for more than half of the province. In 2023, the grain planting area of Jilin Province will reach 5,825,600 hectares, an increase of 40,500 hectares over the previous year. Among them, the rice planting area was 828,800 hectares, a decrease of 4400 hectares; the corn planting area was 4,544,300 hectares, an increase of 74,900 hectares; the soybean planting area was 328,800 hectares, an increase of 18,900 hectares.

Lishu County is a typical black soil area in China, a national key commodity grain production county and corn export base county, and the origin of the 'Lishu model'; Yushu City known as the "world's largest granary", grain output will reach 6.379 billion pounds in 2024; Jiutai District is known as 'the land of fish and rice along the Songhua River', the construction of high-standard farmland to enhance the quality of black soil, the annual output is stable at more than 900 million pounds; Nong'an County grain production was 2.904 million tons, Ranked fourth among the counties (cities) with the highest grain production in the country. From August to October 2024, field visits were carried out in four counties (cities) including Lishu County, Yushu City, Jiutai District and Nong'an County in Jilin Province, The survey was conducted by one-on-one household questionnaire interview. The survey team visited 510 grain-growing households in 72 administrative villages in 24 towns and villages in 4 counties (districts) of Jilin Province. In order to ensure

the representativeness of the samples, the survey selected 4 counties (districts) as sampling areas. A questionnaire survey was carried out by stratified random sampling: in each county (district), towns were divided into three levels: high, middle and low according to the level of economic development, and 1 ~ 6 towns were randomly selected in each layer according to the number of towns, totaling 24 towns; Each township randomly selects 1 ~ 3 villages, and a total of 60 administrative villages are selected; 5 ~ 10 heads of grain farmers were randomly selected in each village. Black soil protection behaviors such as no-tillage, subsoiling, straw returning, and organic fertilizer application are mainly applicable to the corn planting process. Therefore, samples from non-corn growers are eliminated. At the same time, samples with incomplete or non-standard information are eliminated, and 486 valid samples are obtained, with an effective rate of 95.29%.China, and the questionnaires were refined on the basis of interviews with local functionaries and pre-surveys of farmers. These four areas are all key counties (cities) in the typical black soil area that the Jilin provincial government lacks. According to the scale of black soil, the social and economic development and the implementation of specific policies, the typical areas of farmers' black soil protection behavior decision-making in Jilin Province are selected as the research area of this paper.

## 3.2. Data source and sample characteristics

The survey objects are mainly the heads of rural households or family members actually engaged in agricultural production labor. Participants were recruited through online platforms (e.g., Wenjuanxing) and community centers from August 1 to October 31, 2024. Inclusion criteria were: (1) age ≥ 18 years; (2) voluntary participation with signed consent. A total of 510 questionnaires were distributed, and 486 valid responses were collected, yielding a response rate of 95.29%.The sample situation is shown in Table 1. The respondents in the survey area are mainly male, relatively older, and generally have lower education levels. 72% of the farming families were mainly purely agricultural, 59.5 percent operated on land sizes of less than 30 mu, which is a Chinese land area unit, of land or less, 48.4 percent of the sample consisted of farming families with a population of three to four, 47.1 percent had a per capita income of more than RMB¥20,000, and 52.1 percent were engaged in agricultural production for more than 20.

## 3.3. Scale design and descriptive statistics

According to the scale design suggestions of planned behavior theory, combined with the preliminary survey results, and based on the basic information of surveyed farmers, the questionnaire adopts Likert's7-level scale, including reverse, intermediate and positive responses, with corresponding magnitude values ranging from 1 to 7 (strongly disagree as 1, strongly agree as 7), and reverse questions are scored in reverse [58,59]. The variables and their descriptive statistics after unified processing and simplification are shown in Table 2, which shows that the farmers' behavioral and behavioral intentions are generally good, with the proportion of farmers with quantitative values greater than 4 in BI1, BI2, BI3, and BI4 being 63.10%, 70.40%, 65.00%, and 70.10%, respectively; and in BE1, BE2, and BE3, the proportion of farmers with quantitative values greater than 4 being 84.20%, 81.30% and 70.60%.Most of the farmers have behavioral intention and positive black soil conservation behaviors.

## 3.4. Structural equation model

Structural equation modelling (SEM) integrates factor analysis and path analysis methods, which can effectively deal with the structural relationship between variables and overcome the problem of covariance between independent variables, and is a confirmatory method. Structural equation modelling consists of a measurement model (1,2) and a structural model (3), which is usually represented as

$$X = \Lambda x \xi + \delta \tag{1}$$

**Table 1. Descriptive Characteristics of farmers.**

| Variable | Classification criterion | Frequent and Continuous | Proportion (%) |
|---|---|---|---|
| Age | Under 30 | 28 | 5.8 |
| | 31~45 years old | 118 | 24.3 |
| | 46~60years old | 248 | 51 |
| | Over 60 years old | 92 | 18.9 |
| gender | female | 106 | 21.8 |
| | male | 380 | 78.2 |
| education | No schooling | 25 | 5.1 |
| | primary school | 131 | 27 |
| | middle school | 231 | 47.5 |
| | Senior high school \ technical secondary school | 44 | 9.1 |
| | College degree or above | 55 | 11.3 |
| Type of work | Pure agriculture | 350 | 72 |
| | Agriculture and other businesses | 91 | 18.7 |
| | Non-agriculture oriented and concurrently engaged in agriculture | 36 | 7.4 |
| | non-agriculture | 9 | 1.9 |
| Farm size | x≥30 mu | 289 | 59.5 |
| | 30<x≤50 mu | 87 | 17.9 |
| | 50<x≤100 mu | 47 | 9.7 |
| | 100<x≤150 mu | 30 | 6.2 |
| | 150<x≤200 mu | 33 | 6.8 |
| Family population | 1 person | 12 | 2.5 |
| | 2 persons | 117 | 24.1 |
| | 3-4 persons | 235 | 48.4 |
| | More than 5 people | 122 | 25.1 |
| Annual per capita household income | Less than RMB¥10000 | 88 | 18.1 |
| | RMB¥10000~15000 | 90 | 18.5 |
| | RMB¥15001~20000 | 79 | 16.3 |
| | More than RMB¥20000 | 229 | 47.1 |
| Engaged in agricultural production | 1~3 years | 33 | 6.8 |
| | 4~9years | 53 | 10.9 |
| | 10~19years | 147 | 30.2 |
| | More than 20 years | 253 | 52.1 |

$$Y = \Lambda y\eta + \varepsilon \tag{2}$$

Where: $X$ the vector of exogenous observables; Y is the vector of endogenous observables; $\Lambda x$ is the matrix of factor loadings for $\xi$; $\Lambda y$ is the matrix of factor loadings for $\eta$; $\delta$ is the error term for the exogenous indicator $x$; $\varepsilon$ is the error term for the endogenous indicator y. The following table shows the error terms of the exogenous indicators.

$$\eta = \Lambda\eta + \Gamma\delta + \gamma \tag{3}$$

In the equation, $\eta$ is the endogenous latent variable; $\Lambda$ is the relationship between the endogenous latent variables; $\delta$ is the exogenous latent variable; $\Gamma$ is the relationship between the exogenous latent variable and the endogenous latent variable; $\gamma$ is the residual term of the structural equation model.

 

**Table 2. Variable Descriptive statistics.**

| Latent variable | Dimensionality | Observed variable | Mean value | Standard deviation |
|---|---|---|---|---|
| AB | Economic benefit | Satisfaction with economic compensation income ($AB_1$) | 4.846 | 1.178 |
| | Social benefit | Awareness of food security ($AB_2$) | 4.119 | 1.390 |
| | Ecological benefit | Awareness of ecological environmental benefits ($AB_3$) | 4.646 | 1.234 |
| SN | External intention | Friends and relatives claim that it was produced in ($SN_1$) | 4.479 | 1.279 |
| | | The village committee advocates participation in ($SN_2$) | 4.459 | 1.396 |
| | | Villagers advocate participation in ($SN_3$) | 4.348 | 1.288 |
| | | The spreading power of black soil protection ($SN_4$) | 4.023 | 1.403 |
| PBC | Degree of perception | Cognitive degree of black soil development process ($PBC_1$) | 4.556 | 1.167 |
| | Control belief | Awareness of black soil protection measures ($PBC_2$) | 4.490 | 1.155 |
| | | Awareness of capital investment and use of black soil protection ($PBC_3$) | 4.829 | 1.146 |
| BI | | Take the initiative to publicize black soil protection ($BI_1$) | 4.889 | 1.145 |
| | | Willing to continue to protect the black soil ($BI_2$) | 5.021 | 1.106 |
| | | Willing to cooperate with others to carry out black soil protection ($BI_3$) | 4.973 | 1.146 |
| | | Willing to take the initiative to learn the black soil protection measures ($BI_4$) | 4.994 | 1.104 |
| BE | | In the process of using black soil, the relevant protection regulations are strictly observed ($BE_1$) | 5.572 | 1.218 |
| | | Conservation tillage measures such as crop rotation and fallow were adopted in black soil cultivation ($BE_2$) | 5.457 | 1.186 |
| | | Actively participated in black soil conservation projects or training ($BE_3$) | 5.222 | 1.218 |

## 4. Results

### 4.1. Reliability and validity test

In order to ensure the reliability and correctness of the research results, this paper further measures the construction reliability and structural validity of the scale. Cronbach's coefficient is usually used as the index of construction reliability test [60]. As shown in Table 3, the Cronbach's coefficient values for all latent variables are greater than 0.80, indicating that the internal consistency of the latent variables is good.

At the same time, before evaluating the results of structural equation model, confirmatory factor analysis was carried out to test the effectiveness of observed variables on potential variables. According to the model fitness results in Table 4, it can be seen that CMIN/DF (chi-square degree of freedom ratio) =2.828 in the range of 1–3, RMSEA (root mean square error) =0.061, which is within the good range of < 0.08, the other GFI, TLI and CFI test results all reached the excellent level above 0.9. Therefore, synthesizing the results of this analysis can show that the CFA model of farmers' black soil conservation behavior has a good fit.

**Table 3. Variable reliability index value.**

| Variable | Number of terms |
|---|---|
| Attitude toward Behavior (AB) | 3 |
| Subjective Norm (SN) | 4 |
| Perceived Behavioral Control (PBC) | 3 |
| Behavioral Intention (BI) | 4 |
| Behavioral (BE) | 3 |
| Total | 17 |

**Table 4. Model Adaptability Test.**

| Index | Reference standards | Actual measurement results |
|---|---|---|
| CMIN/DF | 1-3 is excellent, 3–5 is good | 2.828 |
| RMSEA | <0.05 is excellent, <0.08 is good | 0.061 |
| GFT | >0.9 is excellent, >0.8 is good | 0.929 |
| TLI | >0.9 is excellent, >0.8 is good | 0.951 |
| CFI | >0.9 is excellent, >0.8 is good | 0.960 |

Under the premise that CFA model of farmers' black soil protection behavior scale has good adaptability, the convergence validity (AVE) and combination reliability (CR) of each dimension of the scale will be further test-ed. The verification process calculates the standardized factor load of each measurement item in the corresponding dimension through the established CFA model. Then, through the calculation formulas of AVE and CR, the convergence validity and combination reliability of each dimension are calculated. According to the standard, the minimum requirement of AVE value is 0.5, and the minimum requirement of CR value is 0.7, which shows that it has good convergence validity and combination reliability. According to the analysis results in Table 5, it can be seen that in the validity test of the farmers' black soil protection behavior scale, AVE values of each dimension are above 0.5 and CR values are above 0.7, which can be summarized to show that all dimensions have good convergence validity and combination reliability.

The validity test mainly observes the discriminative validity among variables, which refers to the low correlation and significant difference between latent variables, and it can be evaluated by comparing the magnitude of the correlation coefficient between the square root of the average variance extraction and the variable. According to the standard proposed by Fornell and Larcker [61], if the correlation coefficient between a variable and other variables is less than the square root of the average variance of the variable, it shows that the variable has good discrimination validity. As

**Table 5. Convergence Validity Test.**

| Path relation | | | Significance estimation | | | | Topic reliability | | Component reliability | Convergence validity |
|---|---|---|---|---|---|---|---|---|---|---|
| | | | UnStd. | S.E. | C.R. | P | Std. | SMC | CR | AVE |
| $AB_3$ | <-- | AB | 1.000 | | | | 0.674 | 0.454 | 0.760 | 0.514 |
| $AB_2$ | <-- | AB | 1.241 | 0.098 | 12.676 | *** | 0.743 | 0.552 | | |
| $AB_1$ | <-- | AB | 1.035 | 0.082 | 12.574 | *** | 0.731 | 0.534 | | |
| $SN_4$ | <-- | SN | 1.000 | | | | 0.607 | 0.368 | 0.850 | 0.592 |
| $SN_3$ | <--- | SN | 0.992 | 0.083 | 11.897 | *** | 0.657 | 0.432 | | |
| $SN_2$ | <--- | SN | 1.453 | 0.101 | 14.436 | *** | 0.887 | 0.787 | | |
| $SN_1$ | <--- | SN | 1.329 | 0.092 | 14.428 | *** | 0.886 | 0.785 | | |
| $PBC_3$ | <--- | PBC | 1.000 | | | | 0.794 | 0.630 | 0.852 | 0.657 |
| $PBC_2$ | <--- | PBC | 1.075 | 0.057 | 18.948 | *** | 0.846 | 0.716 | | |
| $PBC_1$ | <--- | PBC | 1.014 | 0.057 | 17.831 | *** | 0.791 | 0.626 | | |
| $BI_1$ | <--- | BI | 1.000 | | | | 0.778 | 0.605 | 0.910 | 0.718 |
| $BI_2$ | <--- | BI | 1.127 | 0.051 | 22.165 | *** | 0.907 | 0.823 | | |
| $BI_3$ | <--- | BI | 1.133 | 0.053 | 21.432 | *** | 0.881 | 0.776 | | |
| $BI_4$ | <--- | BI | 1.013 | 0.052 | 19.521 | *** | 0.818 | 0.669 | | |
| $BE_1$ | <--- | BE | 1.000 | | | | 0.884 | 0.781 | 0.908 | 0.768 |
| $BE_2$ | <--- | BE | 1.054 | 0.036 | 29.480 | *** | 0.956 | 0.914 | | |
| $BE_3$ | <--- | BE | 0.883 | 0.040 | 21.929 | *** | 0.780 | 0.608 | | |

Note: *** and ** are significant at 1% and 5% levels respectively.

shown in Table 6, the data in bold font in the table is the square root of the squared variance extraction, which is generally larger than all the values in its column. Therefore, the discriminant validity of the measurement model in this study is appropriate [62–64].

## 4.2. SEM results

According to the operation results of SEM (Fig 2, Tables 7 and 8), the research hypotheses H1~H6 are confirmed, indicating that the logic of farmers' black soil protection behavior conforms to TPB theory. Farmers follow the logical path of cognitive judgment – intention selection – behavioral in the process of black soil protection behavior. The three latent variables of farmers' attitudes towards black soil protection behavior, subjective norms, and perceived behavioral control determine their willingness to protect black soil. Farmers' willingness to protect black soil determines their response to black soil protection behavior. Behavioral intention plays a mediating role between their cognition (AB, SN, PBC) and behavioral and perceived behavioral control directly affects behavioral.

The path coefficient from attitude toward behavioral (AB) to behavioral intention (BI) in farmers' black soil conservation behavior is 0.286, which is significant at 1% level. The path coefficients of behavioral attitude and its four observed variables (AB1, AB2, AB3) are 0.73, 0.74, and 0.67, respectively (Fig 2), which include farmers' awareness of the economic, social, and ecological benefits brought by black soil protection, indicating that positive cognition of the benefits of black soil protection can enhance farmers' willingness to participate in black soil protection. The higher benefit of black soil protection is also the main embodiment of the relevant benefits of farmers' participation in black soil protection, which accords with the economic principle. Therefore, the government needs to provide reasonable compensation for black soil protection farmers, and distribute it in a timely manner to stabilize their income. At the same time, black soil protection publicity and training activities should be carried out for farmers who have not carried out black soil protection, so as to improve farmers' cognition level of black soil protection.

Subjective norms are also the main factor affecting behavior intention, and its path coefficient is 0.098, which is significant at the level of 5%. The path coefficients of the subjective norm and its four observed variables (SN1, SN2, SN3, SN4) are 0.89, 0.89, 0.66 and 0.61, respectively (Fig 2), which includes farmers' judgment on the degree of their awakening influenced by the external factors and their perception of the pressure from the outside world, indicating that farmers will be influenced and pressured by relatives and friends and village committees, among which the influence and pressure from relatives and friends and village committees are relatively stronger. As a basic mass autonomous organization, the village committee has a strong influence on the production and life of farmers in the village collective, and also plays an important role in mobilizing farmers' black soil protection behavior. Therefore, when the government promotes black soil protection behavior, it also needs to pay attention to the training of rural cadres, improve their awareness of black soil protection behavior, and better play the role of the village committee.

**Table 6. Differential validity test.**

| | Convergent Validity | Discriminant Validity | | | | |
|---|---|---|---|---|---|---|
| | AVE | BE | BI | PBC | SN | AB |
| BE | 0.768 | 0.876 | | | | |
| BI | 0.718 | 0.564 | 0.847 | | | |
| PBC | 0.657 | 0.423 | 0.596 | 0.811 | | |
| SN | 0.592 | 0.226 | 0.387 | 0.468 | 0.769 | |
| AB | 0.514 | 0.427 | 0.562 | 0.658 | 0.415 | 0.717 |

Note: The bold numbers on the diagonal represent the root of AVE, while the lower triangle represents the Pear-son correlation coefficient of the structural plane.

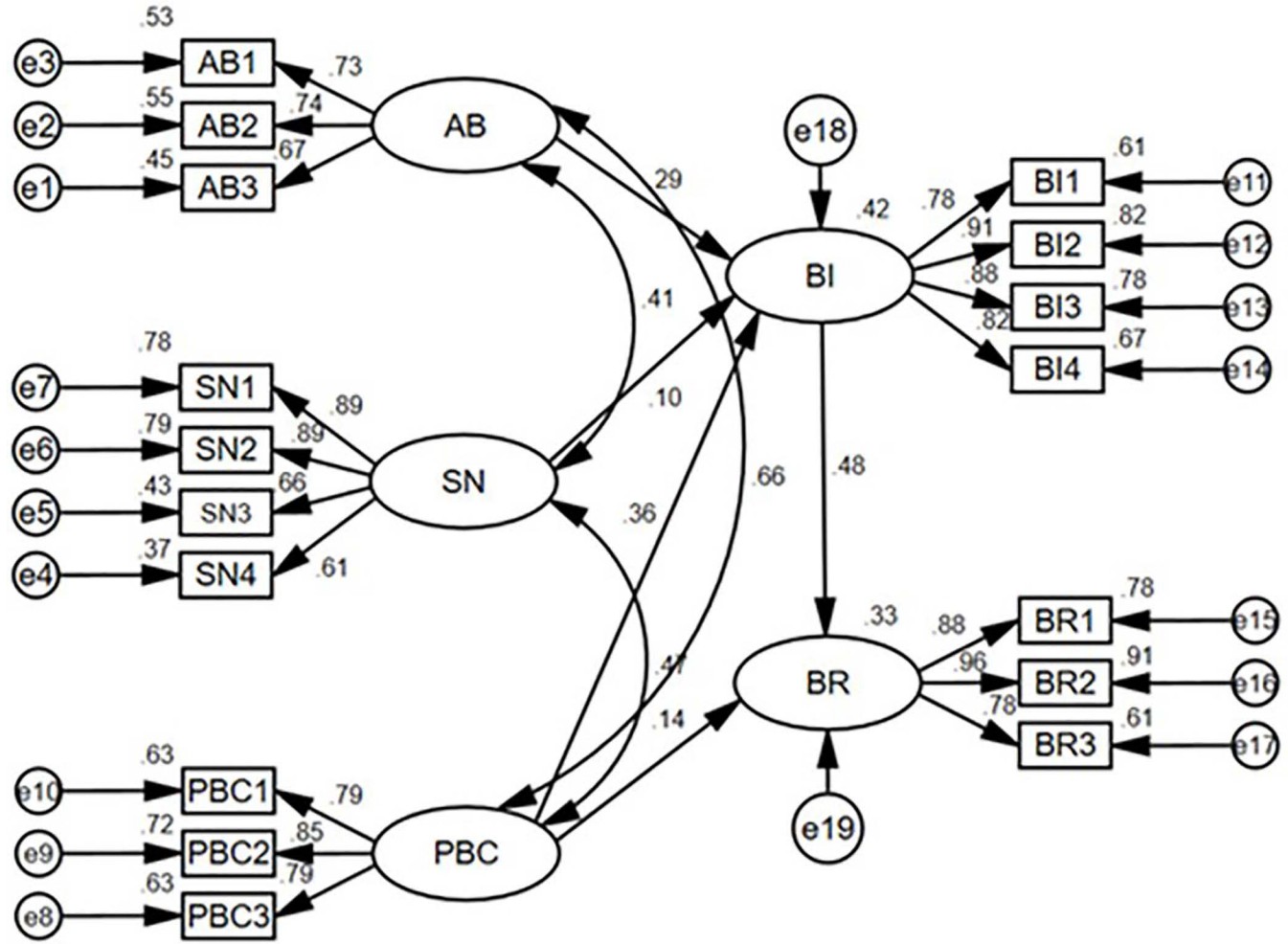

**Fig 2. Structural equation model and estimation results of farmers' black soil protection behavior.** Note: Chi-square value = 313.938 Degree of=111 Chi-square/DF = 2.828 p = 0.000 GFI = 0.929 AGFI = 0.902 RMSEA = 0.061 CFI = 0.960 TLI = 0.951.

**Table 7. Estimation Results of Structural Equation Model.**

| | | | Unstd. | S.E. | C.R. | P | Std.β |
|---|---|---|---|---|---|---|---|
| AB | <--> | PBC | 0.498 | 0.057 | 8.721 | *** | 0.659 |
| AB | <--> | SN | 0.293 | 0.047 | 6.190 | *** | 0.414 |
| SN | <--> | PBC | 0.362 | 0.050 | 7.210 | *** | 0.468 |
| BI | <--- | AB | 0.306 | 0.075 | 4.090 | *** | 0.286 |
| BI | <--- | SN | 0.103 | 0.051 | 2.020 | 0.043 (**) | 0.098 |
| BI | <--- | PBC | 0.355 | 0.067 | 5.273 | *** | 0.362 |
| BE | <--- | PBC | 0.166 | 0.067 | 2.482 | 0.013 (**) | 0.140 |
| BE | <--- | BI | 0.580 | 0.071 | 8.224 | *** | 0.481 |

Note: * * * and * * are significant at 1% and 5% respectively.

**Table 8. Mediating Effect Test.**

| Path | Effect | SE | S.E. | Bias-corrected 95% CI | | P |
|---|---|---|---|---|---|---|
| | | | | Lower | Upper | |
| AB→BI→BE | Direct effect | 0.000 | 0.000 | — | — | — |
| | Indirect effect | 0.138 | 0.041 | 0.070 | 0.230 | 0.001 |
| | Total effect | 0.138 | 0.041 | 0.070 | 0.230 | 0.001 |
| | Direct effect | 0.000 | 0.000 | — | — | — |
| SN→BI→BE | Indirect effect | 0.047 | 0.028 | −0.004 | 0.107 | 0.063 |
| | Total effect | 0.047 | 0.028 | −0.004 | 0.107 | 0.063 |
| | Direct effect | 0.140 | 0.064 | 0.006 | 0.259 | 0.036 |
| PBC→BI→BE | Indirect effect | 0.174 | 0.051 | 0.082 | 0.283 | 0.001 |
| | Total effect | 0.314 | 0.069 | 0.170 | 0.441 | 0.001 |

Note: * * * and * * are significant at 1% and 5% respectively.

Perceptual behavioral control is a factor affecting behavioral intention and a major factor affecting behavioral. The path coefficients of behavior control and its three observed variables (PBC1, PBC2 and PBC3) are 0.79, 0.85 and 0.79, respectively (Fig 2), which include farmers' cognitive level of the process of carrying out black soil protection behavior, as well as their control belief in mastering black soil protection measures and capital in-vestment and use, indicating that farmers' specific cognition of black soil protection behavior not only helps to improve farmers' willingness to participate, but also directly promotes farmers' participation behavior. This reflects that black soil protection involves the vital interests of farmers, and farmers maintain an extremely cautious attitude towards black soil protection. Among them, the path coefficients of farmers' awareness of the process of carrying out black soil conservation (PBC1) and awareness of the investment and use of black soil conservation funds (PBC3) were equally large, indicating that the awareness of the process of carrying out black soil conservation (PBC1) and the awareness of the investment and use of black soil conservation funds (PBC3) were the most important observational variables influencing farmers' behavioral intention and behavioral. The possible reason is that farmers with higher awareness of the development of black soil protection (PBC1) and the in-vestment and use of black soil protection funds (PBC3) generally have a relatively higher education level or have stronger physical influence in agricultural production. In the process of farmers' black soil protection behavior, they are also the easiest group to understand the policy and the most important mobilization object of the government, so they will show their willingness and behavior to participate in black soil protection more actively. However, due to the small operation area of farmers, the attraction of total benefit is not enough to break the preference of many farmers for traditional farming methods, so farmers lack the initiative to adopt conservation farming technology; most farmers are risk-averse and are usually cautious about the application of new technologies such as conservation tillage, and "seeing is believing" is often the main basis for their technology choice [17]. More importantly, at present, China's land management law is very clear about the responsibilities of governments at all levels in land protection, but it does not specify the responsibilities of land contractors for the quality protection of black soil, which leads to a relatively low awareness of farmers' protection measures for black soil (PBC2). Therefore, publicity, training and demonstration should be strengthened to enhance farmers' awareness of black soil conservation farming.

Behavioral intention is the main factor affecting the response, and its total standardized effect on behavioral is the highest at 0.481 (Table 7), which on the one hand can directly contribute to the behavioral of farmers, and on the other hand also plays a mediating effect from farmers' awareness of black soil protection to their behavior. Therefore, farmers' black soil protection behavior should not only pay attention to the protection effect of different technical models, but also pay attention to farmers' willingness, and jointly improve the overall effect of farmers' black soil protection from the perspective of technology and management.

Attitudes towards behavioral, subjective norms and perceived behavioral control are all in significant correlations with each other. In essence, attitudes towards behavioral, subjective norms, perceived behavior control and their observation variables are all influenced by factors such as the gender, age, education level of farmers' decision makers, household population and per capita annual income. For example, the path coefficient of behavioral attitude and perceived behavioral control is 0.659, indicating that farmers' cognition level of economic, social and ecological benefits brought by black soil protection is highly correlated with farmers' cognition level of black soil protection process, protection measures and funds. As mentioned above, farmers with a high level of perceptual behavior control may have a high level of education, or they may be the primary target of government mobilization for black soil protection, so their policy awareness of black soil protection will be at a relatively high level, and thus their awareness of the benefits of black soil protection will be relatively sufficient. The reasons for the correlation among behavioral attitudes, perceived behavioral control and subjective norms are similar. Generally speaking, it is necessary to pay attention to the mutual influence of farmers' cognition, so different policies and measures also need to be effectively coordinated and connected to improve farmers' cognition level of black soil protection policies and promote farmers' participation in black soil protection.

Bootstrap method was used to test the mediating effect of farmers' behavioral intention through 2000 iterations of calculations. The results in Table 8 show that there is no direct effect on path AB→BI→BE, but the indirect effect value is 0.138, the confidence intervals are all positive, excluding 0, and the P value is 0.001; In the path SN→BI→BE, this paper pays special attention to the potential mediation path of variable SN affecting BE through variable BI. According to the analysis results, although the direct effect of this path is 0, indicating that there is no direct correlation between SN and BE, some indirect effect is observed. Specifically, the indirect effect of SN on BE through BI is 0.047. However, the statistical significance of this indirect effect is weak, and the P value is 0.063, which is not only close to the traditional significance level of 0.05. This means that a possible mediating effect may have been detected, but its statistical reliability is not strong. In addition, the 95% confidence interval of mediation effect is [−0.004, 0.107], which contains 0, indicating that the effect value may include the case of no effect. Although the data shows that the variable SN may have a certain influence on BE through the variable BI, it is not certain that this mediating effect really exists in the population because the confidence interval contains 0 and the p value is only close to the significant level; the analysis results of path PBC→BI→BE show that there are not only significant intermediary effects, with an effect value of 0.174 and a P value of 0.001, but also significant direct effects, with an effect value of 0.140 and P value of 0.036. This indicates that the variable PBC not only directly influences BE, but also indirectly influences BE through the variable BI. The total effect value is 0.314, the P value is 0.001, and the confidence interval is [0.170, 0.441], which further proves the importance of this path.

## 5. Conclusions

On the basis of the TPB, this paper analyzes model of farmers' decision-making on black soil protection, and conducts an empirical analysis with the typical regional survey data of farmers' decision-making on black soil protection behavior in Jilin Province. The empirical results indicate that behavior attitude, subjective norms and perceived behavior control reveal the determinants affecting intention and practice of black soil protection. The path coefficient of AB2 (0.74) was the highest in behavior attitude; The SN1 (0.89) and SN2 (0.89) coefficients are equally high in subjective norms; PBC2 (0.85) was higher in perceived behavior control. Therefore, the government should give priority to AB2, SN1, SN2 and PBC2 in cultivating farmers' behavior of black soil protection. What's more, these three determinants are affected by the personal characteristics (gender, age, etc.), family characteristics (population, annual income, etc.), and cultivated land characteristics (area, quality) of the farmers, What mechanisms these three determinants behavioral intention and behavioral response still need further study.

The results of SEM showed that the TPB theory has a strong explanatory power to the black soil protection behavior of farmers, and the logic of the black soil protection behavior of farmers follows the basic path of Cognitive Judgment – Intention Choice – Behavior. The comprehensive influence path coefficients of the three dimensions of farmers' cognition

of black soil protection on their black soil protection behavior are SN (0.098) <AB (0.286) <PBC (0.362) in descending order. When cultivating farmers' black soil protection behavior, the government should first pay attention to the PBC of farmers' black soil protection behavior. In addition, in the mediation effect test, the importance of PBC was once again proved. PBC has not only an indirect effect (0.051) but also a direct effect (0.064) on BE, and the total effect is 0.069 and significant at 1%.

The results of this paper are significant to the policy-making by the central and local governments. In terms of strategy of black soil protection, it is critical to pay attention to not only the improvement and development of black soil protection measures, but also the farmers' cognition, willingness and behavior. In the concrete implementation process, the basic principle of black soil protection is to respect farmers' willingness and increase farmers' income. It is necessary to formulate reasonable compensation criteria and provide them the income compensation. It is essential not only to strengthen publicity and training on black soil protection for farmers to enhance their awareness of protection benefits, but also to systematically train rural cadres to coordinate and promote the policy implementation of black soil protection.

This paper takes farmers in Jilin Province as the research object, and does not distinguish whether farmers plant corn or rice. There are essential differences between them in the way and behavior of black land protection. Some conclusions may change because of the change of crop type, planting scale and geographical environment, and the research of farmers' behavior of black land protection cannot be separated from the study of land property rights system. The core of this paper is to study the relationship among the three. Results must be viewed in the context of research limitations and recommended future research directions. While this study has significantly improved our understanding of black soil conservation by farmers and laid a foundation for further research and practical applications in sustainable conservation black soil development, it is important to recognize its limitations. One limitation concerns the use of Intention as a proxy for the actual behavior, considering that there are studies that indicated a gap between these two variables [65]. Despite this limitation, a substantial body of research, especially within the TPB, demonstrated that intention is a strong predictor of behavior [66]. Therefore, intention is often used as a practical and feasible indicator in black soil conservation research, especially when actual behavioral data is difficult to collect or resource-intensive. Furthermore, the present study can be seen as a critical first step in understanding intention. Future research could build on this, incorporate behavioral measures and examine the transition from intention to action. The study relies on data from farmer household respondents in Jilin Province, China, and the results should be interpreted in this context. Additional testing in different settings will provide further evidence for the generality and robustness of the measurement model.

## Supporting information

**S1 File. Renamed_63c97.**
(XLSX)

## Author contributions

**Conceptualization:** Xu Sun.

**Data curation:** Xu Sun.

**Formal analysis:** Xu Sun.

**Funding acquisition:** Yunxian Yan.

**Investigation:** Rui Yan.

**Methodology:** Xu Sun.

**Project administration:** Yunxian Yan, Ling Zhao.

**Resources:** Ling Zhao.

**Supervision:** Ling Zhao.

**Validation:** Rui Yan.

**Visualization:** Rui Yan.

**Writing – original draft:** Xu Sun.

**Writing – review & editing:** Yunxian Yan.

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
