## [Decision Letter · Decision Letter 0]

7 Jul 2025

Dear Dr. Ling,

Thank you for submitting your manuscript to PLOS ONE. After careful consideration, we feel that it has merit but does not fully meet PLOS ONE’s publication criteria as it currently stands. Therefore, we invite you to submit a revised version of the manuscript that addresses the points raised during the review process.

We look forward to receiving your revised manuscript.

Kind regards,

Ying Ma, Ph.D.

Academic Editor

PLOS ONE

Journal Requirements:

2. You indicated that ethical approval was not necessary for your study. We understand that the framework for ethical oversight requirements for studies of this type may differ depending on the setting and we would appreciate some further clarification regarding your research. Could you please provide further details on why your study is exempt from the need for approval and confirmation from your institutional review board or research ethics committee (e.g., in the form of a letter or email correspondence) that ethics review was not necessary for this study? Please include a copy of the correspondence as an ""Other"" file.

“This work was supported by The Education Department of Jilin Province (JJKH20250609BS).”

5. We note that you have indicated that there are restrictions to data sharing for this study. PLOS only allows data to be available upon request if there are legal or ethical restrictions on sharing data publicly. For more information on unacceptable data access restrictions, please see http://journals.plos.org/plosone/s/data-availability#loc-unacceptable-data-access-restrictions.

6. PLOS requires an ORCID iD for the corresponding author in Editorial Manager on papers submitted after December 6th, 2016. Please ensure that you have an ORCID iD and that it is validated in Editorial Manager. To do this, go to ‘Update my Information’ (in the upper left-hand corner of the main menu), and click on the Fetch/Validate link next to the ORCID field. This will take you to the ORCID site and allow you to create a new iD or authenticate a pre-existing iD in Editorial Manager.

7. We note that Figure 2 in your submission contain map/satellite images which may be copyrighted. All PLOS content is published under the Creative Commons Attribution License (CC BY 4.0), which means that the manuscript, images, and Supporting Information files will be freely available online, and any third party is permitted to access, download, copy, distribute, and use these materials in any way, even commercially, with proper attribution. For these reasons, we cannot publish previously copyrighted maps or satellite images created using proprietary data, such as Google software (Google Maps, Street View, and Earth). For more information, see our copyright guidelines: http://journals.plos.org/plosone/s/licenses-and-copyright.

 1. You may seek permission from the original copyright holder of Figure 2 to publish the content specifically under the CC BY 4.0 license. 

8. Please ensure that you refer to Figure 1 in your text as, if accepted, production will need this reference to link the reader to the figure.

9. We note you have included a table to which you do not refer in the text of your manuscript. Please ensure that you refer to Table 1 in your text; if accepted, production will need this reference to link the reader to the Table.

Reviewers' comments:

Reviewer's Responses to Questions

**Comments to the Author**

1. Is the manuscript technically sound, and do the data support the conclusions?

Reviewer #1: No

Reviewer #2: Yes

2. Has the statistical analysis been performed appropriately and rigorously?

Reviewer #1: No

Reviewer #2: Yes

3. Have the authors made all data underlying the findings in their manuscript fully available?

Reviewer #1: Yes

Reviewer #2: Yes

4. Is the manuscript presented in an intelligible fashion and written in standard English?

Reviewer #1: Yes

Reviewer #2: Yes

Reviewer #1: Determinants of Farmers' Decision-Making Behaviour for Black Soil Protection in Northeast China — A Case of Jilin Province

The topic is relevant because black soil degradation in China is a critical environmental problem with significant implications for food security.

The paper attempts to adapt the Theory of Planned Behaviour (TPB) to analyse farmers' decision-making behaviour.

I suggest some revisions before considering it for publication:

1. The research questions are not clearly formalised: they do not emerge explicitly in either the introduction or the objectives section. It is essential to make them clear, explicit and testable, especially since this is an SEM model.

2. The originality is only partial. The application of TPB to agricultural behaviour is now widespread: there is no strong justification for the innovative contribution compared to the extensive existing literature.

3. The literature is largely self-referential and not up to date with recent contributions on the adoption of sustainable agricultural practices at the international level.

4. The limitations of the TPB are not critically discussed, for example, the lack of attention to the institutional context and structural inequalities.

5. The use of SEM is technically appropriate for testing models with latent variables and multiple relationships. Reliability (Cronbach's α), convergent validity (AVE) and discriminant validity tests are correctly implemented. However, the sampling process is weak and it is not explained transparently how the 486 farmers were selected. Recruitment via online platforms does not guarantee representativeness.

6. The justification for the study area (Jilin) is broad but poorly connected to theoretical or comparative reasoning. There is no reflection on how generalisable the results are outside this context.

7. The SEM formulas are reported correctly, but the text does not clearly illustrate how the standardised weights were interpreted.

8. The results are consistent with the TPB literature: behavioural intention mediates the effect of cognitive and normative variables on actual behaviour. However, the tables are numerous and redundant. The information on AVE, CR, and CFI could be better summarised. The figures are descriptive and do not add interpretative substance to the model.

9. The standardised estimates report limited effects: for example, the contribution of subjective norms is statistically weak (β=0.098).

10. There is a lack of sensitivity analysis or robustness testing of the model.

11. The discussion highlights the crucial role of training, institutional communication and economic incentives in promoting protective behaviour. However, this remains too general and lacks a real comparison with international literature or similar contexts (e.g. other countries with soil degradation problems).

12. Policy implications are stated but not supported by concrete scenarios or simulations.

13. Methodological limitations are only mentioned marginally (absence of causality, risk of bias in questionnaires, etc.), and future research developments are vague.

Good work

Reviewer #2: I would like to thank the journal's editorial team.

The submitted manuscript has valuable contributions but requires some minor revisions before it can be accepted. Therefore, I would like to draw the authors' attention to the following points:

Comment 1: Remove Cronbach’s alpha from Table 3.

Comment 2: The lack of comparative analysis with findings from other related studies. It is necessary to discuss research that has obtained similar or contrasting results and explain the reasons for these similarities and differences.

Comment 3: The following articles should be cited in the paper:

Rastegari, H., Nooripoor, M., Sharifzadeh, M., & Petrescu, D. C. (2023). Drivers and barriers in farmers’ adoption of vermicomposting as keys for sustainable agricultural waste management. *International Journal of Agricultural Sustainability, 21*(1), 2230826.

Petrescu-Mag, R. M., Rastegari, H., Hartel, T., Reti, K. O., & Petrescu, D. C. (2025). Biocultural tourist experience in Romania’s High Nature Value rural landscape: Application of an extended Theory of Planned Behavior. *PLoS One, 20*(5), e0324444.

**Do you want your identity to be public for this peer review?** For information about this choice, including consent withdrawal, please see our Privacy Policy

Reviewer #1: No

Reviewer #2: **Yes: ** Hamid Rastegari

---

## [Author Response · Author response to Decision Letter 1]

2 Oct 2025

Response to the Reviewers’ Comments

We thank the reviewers for taking the time to read and give their comprehensive and constructive comments, which have improved our manuscript. Below, we provide a point-by-point response to your comments and suggestions and how each one has been addressed in the revision.

Response to Reviewer 1 Comments

Comment 1. The research questions are not clearly formalised: they do not emerge explicitly in either the introduction or the objectives section. It is essential to make them clear, explicit and testable, especially since this is an SEM model.

Response: We are sorry for the shortcomings in our previous manuscript. The bibliography recommended by reviewers is of great help to the manuscript and helps the author to have a deeper understanding of the existing research. In the abstract part, we have made some modifications, and at the same time, we have written the clear research questions of this paper in the penultimate paragraph of the introduction.

Comment 2. The originality is only partial. The application of TPB to agricultural behaviour is now widespread: there is no strong justification for the innovative contribution compared to the extensive existing literature.

Response: We thank the reviewer for this comment. We have made relevant modifications in the penultimate paragraph of the introduction of the article, which fully illustrates the innovation of this research.

Comment 3. The literature is largely self-referential and not up to date with recent contributions on the adoption of sustainable agricultural practices at the international level.

Response: We thank the reviewer for this comment. In the third paragraph of our introduction, we added sustainable practices for black soil conservation in the United States and Ukraine.

Comment 4. The limitations of the TPB are not critically discussed, for example, the lack of attention to the institutional context and structural inequalities.

Response: We thank the reviewer for this comment. In the penultimate paragraph of the introduction, we have given a detailed answer to this question.

Comment 5. The use of SEM is technically appropriate for testing models with latent variables and multiple relationships. Reliability (Cronbach's α), convergent validity (AVE) and discriminant validity tests are correctly implemented. However, the sampling process is weak and it is not explained transparently how the 486 farmers were selected. Recruitment via online platforms does not guarantee representativeness.

Response: First of all, I would like to thank the reviewers for their approval of the model, and thank you for your questions. In response to your question, we detailed the extraction of the survey in the third paragraph of Materials and Methods in the article.

Comment 6. The justification for the study area (Jilin) is broad but poorly connected to theoretical or comparative reasoning. There is no reflection on how generalisable the results are outside this context.

Response: We thank the reviewer for this comment. In the third paragraph of Materials and Methods in the article, we focused on the reasons for selecting Jilin Province, because Jilin Province in China is a typical area for black soil protection, and the "pear tree model" is also promoted throughout the country.

Comment 7. The SEM formulas are reported correctly, but the text does not clearly illustrate how the standardised weights were interpreted.

Response: We thank the reviewer for this comment. We illustrate the means and standard deviations of the selected variables in Table 2, and verify the reliability of the scale in the first paragraph of the results, which indicate good internal consistency. So no normalized weights are reported.

Comment 8. The results are consistent with the TPB literature: behavioural intention mediates the effect of cognitive and normative variables on actual behaviour. However, the tables are numerous and redundant. The information on AVE, CR, and CFI could be better summarised. The figures are descriptive and do not add interpretative substance to the model.

Response: We thank the reviewer for these comments and affirmation of the manuscript. What we want to explain is that the AVE, CR, and CFI you mentioned are not in the same table. In the structural equation model, CFI measures the fitting degree of the CFA model. AVE and CR will be calculated only on the premise that the fitting degree passes. AVE and CR test the convergence validity and combined reliability of each dimension of the scale, so we did not put the three indicators you mentioned in one table. Thank you for your understanding.

Comment 9. The standardised estimates report limited effects: for example, the contribution of subjective norms is statistically weak (β=0.098).

Response: We thank the reviewer for this comment. We explain the reasons for the weak significance of the subjective norm in the third paragraph in the SEM results and refer to three articles of literature.

Comment 10. There is a lack of sensitivity analysis or robustness testing of the model.

Response: We thank the reviewer for this comment. We explain this in the last paragraph of our conclusion in the article, thank you for your understanding.

Comment 11. The discussion highlights the crucial role of training, institutional communication and economic incentives in promoting protective behaviour. However, this remains too general and lacks a real comparison with international literature or similar contexts (e.g. other countries with soil degradation problems).

Response: We thank the reviewer for this comment. In the third paragraph of the introductory part of the article, we introduce the practices of black land protection in the United States and Ukraine.

Comment 12. Policy implications are stated but not supported by concrete scenarios or simulations.

Response: We thank the reviewer for this comment. We have made corresponding changes in the conclusions

Comment 13. Methodological limitations are only mentioned marginally (absence of causality, risk of bias in questionnaires, etc.), and future research developments are vague.

Response: We thank the reviewer for this comment. We revised this in the conclusion part, and put forward the future research development.

Response to Reviewer 2 Comments

Comment 1: Remove Cronbach’s alpha from Table 3.

Response: We thank the reviewer for this comment. Changes have been made as you suggested.

Comment 2: The lack of comparative analysis with findings from other related studies. It is necessary to discuss research that has obtained similar or contrasting results and explain the reasons for these similarities and differences.

Response: We thank the reviewer for this comment. According to your question, we explain the reason why the subjective norm is weakly significant in the third paragraph of SEM results, and refer to three literatures.

Comment 3: The following articles should be cited in the paper:

Response: We thank the reviewer for this comment. We have already referenced both articles in our article.

---

## [Decision Letter · Decision Letter 1]

27 Oct 2025

Dear Dr.  Ling,

Thank you for submitting your manuscript to PLOS ONE. After careful consideration, we feel that it has merit but does not fully meet PLOS ONE’s publication criteria as it currently stands. Therefore, we invite you to submit a revised version of the manuscript that addresses the points raised during the review process.

We look forward to receiving your revised manuscript.

Kind regards,

Ying Ma, Ph.D.

Academic Editor

PLOS ONE

Journal Requirements:

Reviewer's Responses to Questions

**Comments to the Author**

Reviewer #1: (No Response)

Reviewer #2: All comments have been addressed

2. Is the manuscript technically sound, and do the data support the conclusions?

Reviewer #1: Yes

Reviewer #2: Yes

3. Has the statistical analysis been performed appropriately and rigorously?

Reviewer #1: Yes

Reviewer #2: Yes

4. Have the authors made all data underlying the findings in their manuscript fully available?

Reviewer #1: Yes

Reviewer #2: Yes

5. Is the manuscript presented in an intelligible fashion and written in standard English?

Reviewer #1: Yes

Reviewer #2: Yes

Reviewer #1: The manuscript has been substantially improved, all major reviewer comments have been properly addressed, and the scientific contribution is now clear and robust. Only minor editorial refinements are needed before acceptance. I suggest a few minor revisions:

In the introduction (lines 28–41), write, ‘As one of the three black soil areas in the northern hemisphere, north-eastern China is an important area for grain production... It takes 200–400 years to generate 1 cm of black soil’. I recommend separating the geographical part from the agronomic part.

Lines 39-48: “However, due to long-term intensive use... improve overall food production capacity”. Here, three problems (“thin-lean-hard”) are described: a bulleted list or shorter sentences would be better.

In Discussion/Interpretative results, lines 349–369: “Perceptual control of behaviour is a factor influencing behavioural intention... awareness of investment and use of funds for black soil protection (PBC3)”. Too dense: a summary and the use of specific references are suggested.

Lines 382–389: “In general, attention should be paid to the mutual influence of farmers” cognition ... promoting farmers' participation in black soil protection.' This repeats the concept already expressed above (covariance of constructs).

I recommend making the narrative more concise by reducing the descriptive geography, clarifying the novelty of applying TPB to black soil protection, and slightly expanding the limits (cross-causality, bias in self-reported practices).

Reviewer #2: (No Response)

**Do you want your identity to be public for this peer review?** For information about this choice, including consent withdrawal, please see our Privacy Policy

Reviewer #1: **Yes: ** Giuseppe Timpanaro

Reviewer #2: No

---

## [Author Response · Author response to Decision Letter 2]

30 Nov 2025

Response to the Reviewers’ Comments

We thank the reviewers for taking the time to read and give their comprehensive and constructive comments, which have improved our manuscript. Below, we provide a point-by-point response to your comments and suggestions and how each one has been addressed in the revision.

Response to Reviewer 1 Comments

Comment 1. In the introduction (lines 28-41), write, 'As one of the three black soil areas in the northern hemisphere, north-eastern China is an important area for grain production… It takes 200-400 years to generate 1 cm of black soil’. I recommend separating the geographical part from the agronomic part.

Response: We have made the changes as you suggested and thank you very much for your suggestions. We didn't separate before because we wanted to explain the importance of black soil in China. After all, China is a big agricultural country. At the same time, food production and land are inseparable.

Comment 2. Lines 39-48: "However, due to long-term intensive use... improve overall food production capacity". Here, three problems ("thin-lean-hard") are described: a bulleted list or shorter sentences would be better.

Response: We thank the reviewer for this comment. We have not revised your modification suggestion. First of all, we have briefly introduced it when explaining "thin-lean-hard", and each sentence cites the corresponding literature. The reason why we didn't make a table is that the current situation of black soil has undergone long-term changes, and not all these data are public. We can only find out the current situation of black soil. Please understand.

Comment 3. In Discussion/Interpretative results, lines 349-369: "Perceptual control of behaviour is a factor influencing behavioural intention... awareness of investment and use of funds for black soil protection (PBC3)". Too dense: a summary and the use of specific references are suggested.

Response: We thank the reviewer for this comment. We have revised your suggestion, deleted some contents, and cited relevant literature.

Comment 4. Lines 382-389: "in general, attention should be paid to the mutual influence of farmers" cognition. promoting farmers' participation in black soil protection.’ This repeats the concept already expressed above (covariance of constructs).

Response: We thank the reviewer for this comment. According to your suggestion, we have made modifications to supplement the novelty of the planned behavior theory for farmers' black soil protection behavior.

---

## [Editor Report · Decision Letter 2]

2 Dec 2025

Determinants of Farmers' Decision-Making Behavior for Black Soil Protection in Northeast China --- A Case of Jilin Province

PONE-D-25-21413R2

Dear Dr. Ling,

We’re pleased to inform you that your manuscript has been judged scientifically suitable for publication and will be formally accepted for publication once it meets all outstanding technical requirements.

Kind regards,

Ying Ma, Ph.D.

Academic Editor

PLOS ONE
---

## [Editor Report · Acceptance letter]

PONE-D-25-21413R2

PLOS One

Dear Dr. Ling,

I'm pleased to inform you that your manuscript has been deemed suitable for publication in PLOS One. Congratulations! Your manuscript is now being handed over to our production team.

Kind regards,

on behalf of

Dr. Ying Ma

Academic Editor

PLOS One